# Resilient performance in healthcare during the COVID-19 pandemic (ResCOV): study protocol for a multilevel grounded theory study on adaptations, working conditions, ethics and patient safety

Petronella Bjurling-Sjöberg [1,2] Camilla Göras [3,4] Malin Lohela-Karlsson [1,5] Lena Nordgren [1,2] Ann-Sofie Källberg [4,6] Markus Castegren,[2,7] Emelie Condén Mellgren [5] Mats Holmberg [2,8] Mirjam Ekstedt [8,9]

For numbered affiliations see end of article.

**Correspondence to**
Dr Petronella Bjurling-Sjöberg;
Petronella.bjurling-sjoberg@ pubcare.uu.se

## ABSTRACT

**Introduction** Since early 2020, the COVID-19 pandemic has challenged societies and revealed the built-in fragility and dependencies in complex adaptive systems, such as healthcare. The pandemic has placed healthcare providers and systems under unprecedented amounts of strain with potential consequences that have not yet been fully elucidated. This multilevel project aims to explore resilient performance with the purpose of improving the understanding of how healthcare has adapted during the pandemic's rampage, the processes involved and the consequences on working conditions, ethics and patient safety.

**Methods** An emerging explorative multilevel design based on grounded theory methodology is applied. Open and theoretical sampling is performed. Empirical data are gathered over time from written narratives and qualitative interviews with staff with different positions in healthcare organisations in two Swedish regions. The participants' first-person stories are complemented with data from the healthcare organisations' internal documents and national and international official documents.

**Analysis** Experiences and expressions of resilient performance at different system levels and times, existing influencing risk and success factors at the microlevels, mesolevels and macrolevels and inter-relationships and consequences in different healthcare contexts, are explored using constant comparative analysis. Finally, the data are complemented with the current literature to develop a substantive theory of resilient performance during the pandemic.

**Ethics and dissemination** This project is ethically approved and recognises the ongoing strain on the healthcare system when gathering data. The ongoing pandemic provides unique possibilities to study system-wide adaptive capacity across different system levels and times, which can create an important basis for designing interventions focusing on preparedness to manage current and future challenges in healthcare. Feedback is provided to the settings to enable pressing improvements.

## Strengths and limitations of this study

► The first project that uses grounded theory to study working conditions, ethics and patient safety during a healthcare crisis through a lens of resilience.

► Using an exploratory multilevel design with openness for the main concern of the participants, can generate in-depth knowledge to understand better the nature and context of adaptations, the interplay between different system levels, the success factors for resilient performance, and the influencing factors and consequences.

► The project enables the development of theoretical understanding from empirical data, and the findings can create an important basis for designing interventions with a focus on preparedness to manage current and future crises and challenges in healthcare.

► Empirical data are gathered over time from staff from different professions, positions and context in healthcare organisations in two mid-size Swedish regions, which has to be considered when interpreting the findings.

► The first-hand perspective of other stakeholders, such as patients and their families, authorities, financiers and politicians, are not included, which should be further explored in future studies.

The findings will also be disseminated through scientific journals and conferences.

## INTRODUCTION

Since early 2020, the COVID-19 pandemic has challenged individuals, healthcare systems and governments, which has revealed the built-in fragility and dependencies in the complex system that constitute current

societies.[1] The response to the threat has varied among nations, but overall, the gravity of the situation was initially underestimated.[2–4] In Sweden, the first COVID-19 cases, without connection to travellers, were diagnosed in early March 2020.[5] Calculations regarding progress, including potential effects on healthcare, were soon surpassed as the pandemic hit some regions substantially earlier and harder than predicted, and the need for hospital care increased exponentially. Hence, healthcare organisations rapidly had to escalate the capacity of care to an unprecedented extent. This factor, together with initially limited knowledge about the disease and appropriate treatment, created ad hoc solutions and uncertainty at all levels. In July, the first wave of the pandemic ebbed out, and healthcare could return to relatively ordinary conditions. In autumn, a second wave hit the nation. Additionally, in the spring of 2021, extensive vaccination efforts with unstable vaccine deliveries were organised and manned. Over the past year, the pandemic has forced healthcare organisations and involved staff to be on alert and to frequently adapt to changing conditions. Additionally, at the time of writing, mutations of the virus created a third wave, contributing to further uncertainty about the future.

Although healthcare professionals around the world are used to encountering crises and events, the pandemic has forced them to work under extremely stressful conditions, which has challenged their performance and ability to provide safe care. The severity of the disease and the high in-flow of patients have also meant prioritising and potentially ethical dilemmas and moral distress. In addition to ethical decisions and facing mortalities far beyond those of previous experience, additional related stressors may include patient overflow, additional shifts, altered medical interventions, insufficient access to protective equipment and intervening outside the area of expertise.[6 7] Depressive symptoms, anxiety, psychological distress, poor sleep quality[8] and post-traumatic stress disorder[9] have been reported by healthcare professionals. Significant strain on the healthcare system entails an increased risk for diagnostic errors, adverse events and delayed care for patients with COVID-19 as well as other illnesses.[10–12] A Swedish medical record review[13] revealed that 18.1% of COVID-19 patients were affected by adverse events during March 2020 to June 2020. This level is twice as high as the frequency for other patients during the same period. Patients admitted to intensive care units were exposed to a four times higher risk of injury.[13] However, experiences and effects of the pandemic are still being elucidated, and knowledge of the processes involved in maintaining safe performance and what support healthcare professionals need in such mega-events is deficient.[14] To generate this additional knowledge, the complexity, interactions and dynamic adaptations that have been made in current societies and healthcare systems need to be considered.[15]

Complex adaptive systems, such as healthcare, are open and constantly interact with their environment. They also contain many elements at different levels that interact in dynamic processes over time.[1 16 17] The microlevel of the system, where the patient's urgent needs require rapid decisions, sometimes operates at different temporalities than the mesolevels and macrolevels, from which the system's resources and management are controlled and regulated. This issue challenges the ability to anticipate and to understand the larger view of the system and to learn from macro-outcomes in addition to the micro.[18] The ability of a complex adaptive system to self-organise makes it resilient to most disturbances, but changing dynamics also have the potential to destabilise the system.[1] To understand work and successful responses to challenges and problems in complex adaptive systems, resilience and explicit resilient healthcare have received increased attention.[19 20]

Resilience requires that an organisation have the 'potential' for resilient performance, that is, the capacity to act in specific ways under certain conditions.[19] Resilience in healthcare is defined as 'the capacity to adapt to challenges and changes at different system levels, to maintain high-quality care'.[21] This perspective involves a proactive approach, which requires the potential to anticipate, to monitor, to learn and to respond.[19] However, resilience is not normative, as adaptations also have the potential for negative consequences.[20] Safety science warns that reducing resilient performance to individuals' capacity to face challenges and complexity may lead to the creation of safety strategies that rely primarily on individuals, and not on system safety.[22 23]

Today, patient safety is often assessed through analysis and the quantification of deviations. By accepting that healthcare is a complex adaptive system and adopting a system perspective with a focus on preventing and strengthening success, a more proactive approach emerges.[24] Such an approach may improve patient safety and quality of care, prevent worker ill health and achieve good working conditions. However, although there has been an increased focus on proactive safety work, success factors regarding working conditions and patient safety are still relatively unexplored.[24 25] Additionally, despite the importance of learning, success factors for adaptive capacity are relatively unexplored,[26] and knowledge about how resilient performance is created is still deficient.[14 15 24 27] Except for some notable ongoing studies,[28 29] empirical studies on resilience have focused mainly on the microlevel.[30 31] To extend the understanding of resilient performance and to explain important interactions between different system levels in healthcare, there is a need to contribute to the theory of resilient healthcare with empirical data from a multilevel perspective.[20 31]

The pandemic has placed healthcare providers and systems under unprecedented strain, with potential consequences on working conditions, ethics and patient safety that have not yet been fully elucidated. The adaptive capacity of a complex adaptive system under extreme pressure can never be fully understood. However, the ongoing pandemic provides unique possibilities to study system-wide adaptive capacity in real time. Applying a

multilevel approach and studying how professionals at different levels of the system adapt their operations under the ongoing pandemic will increase the understanding of how adaptive processes interplay across care contexts and system levels to sustain high-quality performance and safety under extraordinary conditions. This approach will also increase knowledge about useful adaptations and success factors for how resilience is created through rapid decisions and adaptations at the microlevel of the system and in the interplay between the microlevel and mesolevel. The staff's experiences of ethical stress and workload will also be explored. Thus, empirical exploration of resilient performance, influencing factors and consequences across different system levels and times can create an important basis for designing interventions focusing on preparedness to manage current and future challenges in healthcare.

## AIMS AND OBJECTIVES

This study protocol describes the research project 'Resilient performance in healthcare during the COVID-19 pandemic'. The overall aim of this multilevel project is to improve the understanding of how healthcare adapted during the pandemic's rampage, the processes involved, and the consequences on working conditions, ethics and patient safety. The specific objectives and research questions that guide the project are as follows:

1. To explore and describe how staff from different professions and positions in healthcare organisations experienced and managed the pandemic's rampage and thus influenced factors and consequences.

► What were the main challenges for staff at different levels in the healthcare organisations during the initial phase of the pandemic?
► How were the challenges addressed and coped with in different types of care contexts (ie, intensive care, ambulance care, emergency care and specific COVID-19 units)?
► How did the main challenges and responses change over time during the continuing rampage of the pandemic?
► What conditions (internal and external) influenced the responses in different care contexts over time, and with what consequences?

2. To explore and identify expressions of resilient performance at different system levels and times, influencing risk and success factors at the micro-meso-macrolevels, the interrelationships and consequences on sustainable working conditions, and ethics and patient safety in different healthcare contexts.

► How did capacity to monitor, anticipate and adapt in response to changing work demands, ethical and patient safety challenges develop over time in different contexts at microlevel and at mesolevel?
► What risk and success factors on different levels influenced the microlevels and mesolevels response?

► What lessons were learnt, and what potential improvements were implemented?
► How did adaptive processes interplay across care contexts and system levels (micro-meso-macro), and with what consequences?

3. To develop a substantive theory to provide a comprehensive contextualised explanation of resilient performance within healthcare during the pandemic.

► What were the overall triggers that activated capabilities to respond?
► What were the overall responses in terms of adaptive actions, interactions and reactions to monitor, anticipate, prepare and learn over time?
► What conditions and processes supported or undermined successful responses, and how did different system levels interplay?
► What was the outcome in terms of consequences on working conditions, ethics and patient safety?

## METHODS AND ANALYSIS
### Overall design

To explore resilient performance during the COVID-19 pandemic's rampage, this project has an emerging explorative multilevel design based on the grounded theory methodology[32 33] (see figure 1). This type of design is ideal when the aim is to understand phenomena in a research area where previous knowledge is limited and where the concerns of people involved are in focus. The project started in September 2020. To cover the first, second and third wave of COVID-19, including the aftermath, data gathering from staff in the included healthcare organisations is estimated to continue until December 2022. The end of the project is however depending on when theoretical saturation is achieved, that is, when sufficiently comprehensive data to thoroughly understand the characteristics of the phenomenon and process is obtained.[32 33] To address the overall aim and the three study objectives, the research will be conducted in four iterative, flexible and closely integrated cycles of activities, including staff from different professions and levels in healthcare organisations, complemented with data from different documents, as specified below.

Cycle I, microlevel: To explore how healthcare professionals experienced and managed the pandemic's rampage, empirical data from professionals working with COVID-19 patients in different contexts, such as emergency departments, specific COVID-19 units and intensive care, will be gathered. The datasets will be analysed separately and focus on each profession at the time. Interactions with nurse managers and medical directors, as well as interprofessional interactions, influencing factors and consequences in different phases of the pandemic, will also be analysed in the first cycle.

Cycle II, mesolevel: Empirical data from the staff in positions not directly involved in patient care, such as managers at the mesolevel and service departments, etc, will be gathered and analysed; the focus will be on their experiences, that is, interorganisational interactions,

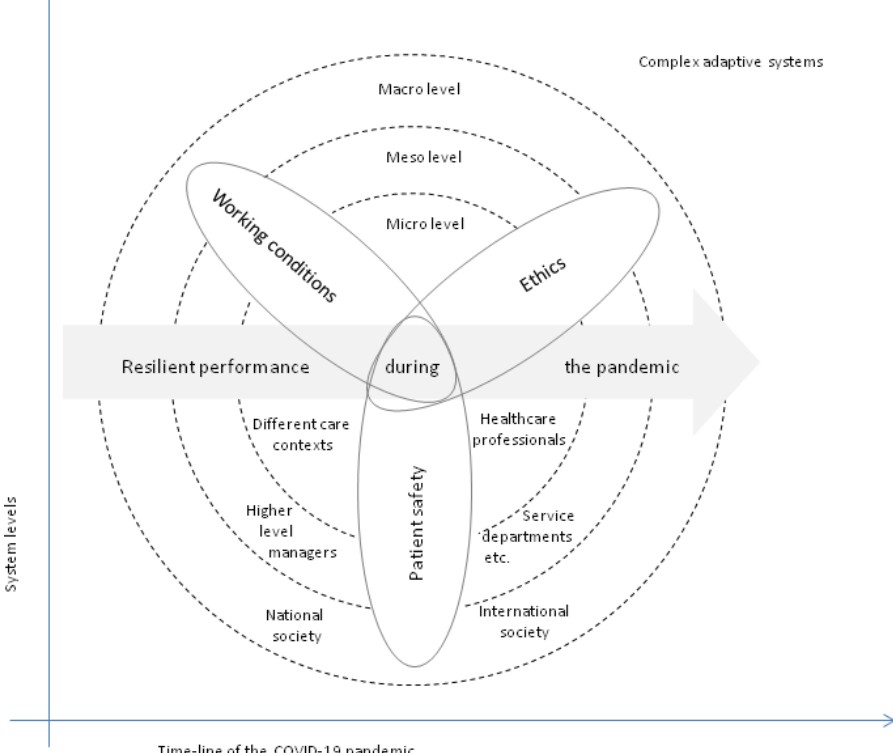

**Figure 1** Overview of the project. Illustration of the micro-meso-and macro system levels, time-line and research focus.

decisions and adaptations, influencing factors and the consequences on microlevel and mesolevel performance at different phases of the pandemic.

Cycle III, multilevel (micro-macro): Empirical data from key informants in context indirectly affected by the pandemic, temporary disaster management of healthcare organisations, internal documents from healthcare organisations and official national and international documents will be gathered and integrated with the findings from cycles I and II. These integrated data will be analysed cross-sectionally and consider the different system levels and phases of the pandemic by focusing on resilient performance, the influencing factors and the consequences in each of the areas of specific interest, namely, sustainable working conditions, ethics and patient safety.

Cycle IV, multilevel (micro-macro): Finally, we will develop a comprehensive contextualised explanation of resilient performance within healthcare during the pandemic. The findings from cycles I–III will be synthesised and supplemented with knowledge from the current literature to provide an empirical and theoretical foundation for the development of a substantive theory of resilient performance during the pandemic rampage.

### Study context
Swedish public healthcare provide the setting for the project. Healthcare in Sweden is largely publicly funded. National regulations provide a foundation for the healthcare system, but the management is self-governed by the 21 regional councils. Therefore, each region is responsible for the organisation and prioritisation of healthcare

resources.[34] The clinical healthcare professionals mainly include registered nurses, some who are specialised within certain fields, for example, intensive care and anaesthesiology; undergraduate assistant nurses; physicians specialised within certain fields, such as medicine, surgery or anaesthesiology, and those under training; and some allied healthcare professions, such as physiotherapists, occupational therapists, dieticians and counsellors. Unlike some other countries, drugs and mechanical ventilation are handled within the healthcare team without pharmacists or respiratory therapists.

The project is performed in the healthcare organisations of two Swedish regions, including seven somatic hospitals and outpatient care. The regions each have a population of ~300 000, which is approximately the median size for the nation. The total number of in-hospital beds in each region is 500–600, with 70–300 beds/hospital, which is a common size of local or county hospitals in the nation.

### Sample and participants
Open sampling and theoretical sampling[33] are performed with the main data source being staff from the included healthcare organisations (microlevels and mesolevels). This empirical data are complemented with data from applicable documents (microlevels-mesolevels-macrolevels), as specified below. Recruitment started September 2020 and is ongoing. Potential participants are contacted through the organisations' email systems. They are informed about the study, aim, procedure and voluntariness and asked about participation.

Initially, aiming to maximise variations in experiences in cycles I–III, open sampling[33] was conducted by recruiting staff from different professions, positions and contexts in healthcare organisations. First, healthcare professionals (registered nurses, assistant nurses, physicians, allied healthcare professions and clinical managers) were recruited who, on a regular or temporary basis, were on duty in the sectors of healthcare that were most affected by the first wave of the pandemic, including emergency departments, specific COVID-19 units, intensive care and operating room departments. Second, higher-level managers were recruited from all parts of the hospitals and primary care, key informants from selected service and support departments, and top-level management in the healthcare organisations and temporary disaster management. At the time of this writing, March 2021, just over 200 participants have been included.

Later, guided by the simultaneously performed analysis in cycle I–IV, theoretical sampling[33] take place, and data are gathered to explore differences over time, to fill gaps in understanding and to saturate emerging concepts and categories. In this phase, we return to already recruited participants for complementary information and recruit additional participants from the healthcare organisations if needed. Additionally, data are used from applicable internal documents from the included healthcare organisations (microlevels and mesolevels), such as statistics, guidelines, protocols and applicable official national and international documents (macrolevel), such as statistics, information, guidelines, recommendations and legal regulations. The inclusion of data from these documents aims to provide enhanced understanding of different factors at the microlevels-mesolevels-macrolevels that influenced the conditions in the included healthcare organisations and thereby had potential consequences on working conditions, ethics and patient safety. Thereby inclusion of documents is guided by the analysis of the participants' first-person experiences.

### Data collection

Primary data consist of the participants' first-person written or verbal narratives, which are supplemented with data from documents as specified above. Data collection started simultaneously with recruitment (September 2020) and will continue until theoretical saturation[32 33] is achieved in all parts of the project. Individuals in the included healthcare organisations who consent to participate are given the choice to tell their story in writing or to participate in individual qualitative interviews. They are also asked to fill out a questionnaire for some demographic data (gender, age, occupation and years in this occupation), regular workplace, role and duties and workplace, role and duties during the pandemic. Additionally, participants are asked to attach supplementary documents from their workplace that they think could facilitate the researchers understanding of their story and to provide contact information if they consent to be contacted again for further questions.

The participants are encouraged to tell their story about their experiences, thoughts and feelings during the pandemic openly. To aid the written narrative, a study-specific guide has been created based on a review of publications and the researchers' experiences. The face validity of the guide was tested by six healthcare professionals, and after minor revisions it was assessed as valid. The guide includes some guiding questions for the topics of specific interest (working conditions, ethics, patient safety, adaptations, influencing factors, consequences and lessons learnt) but urges the participants to tell or to leave out whatever they want.

The interviews are performed by the researchers with great openness for what the informant thinks is important to illuminate, starting with the question 'Can you please tell me how the situation been during the pandemic?' and continuing with illuminating probing questions. A semi-structured interview guide has been developed, which includes the same topics as those found in the guide for the written narratives; however, this guide is used in a flexible manner to ensure that the participants' concerns are captured. Based on the ongoing analysis, both guides are successively refined to support exploration of emerging understandings. All interviews are digitally recorded and transcribed verbatim.

Data collection from documents on different system levels and periods of time, as described above, are conducted first through the participants who provide internal documents from their workplace (microlevels and mesolevels). Second, based on needs that emerge in the analysis, the researchers request useful official documents from the included healthcare organisations (microlevels and mesolevels) and applicable authorities (macrolevel) or collect them from their websites if available.

### Data management and analysis

Demographic data are managed in SPSS Statistics 22[35] to provide descriptive statistics of the participants. All qualitative data are managed in NVivo[36] and analysed using constant comparative analysis according to the principles of grounded theory.[33] The research questions guide the analysis but are open and modifiable to what emerge in data as significant. The analysis process started with the first data collection and will continue until theoretical saturation is achieved in all parts of the project.

Throughout the analysis process, data and emerging concepts and categories are compared back and forth, moving between open, axial and selective coding as new data are collected and the theorising proceeds. The theorising includes reflecting, writing memos and drawing diagrams of interpretations, ideas, assumed associations and theoretical reflections related to each emerging category, moving towards explanation in an abductive approach,[37] shifting between inductive interpretation and deductive testing as hypotheses evolve. Tools, such as the 'paradigm model' to facilitate analysis of process and the 'contextual matrix' to facilitate analysis of the

different system levels,[33] are used in a flexible manner to facilitate accuracy without restricting the creativity of the analysis. Finally, the developed theory are validated in the original empirical raw data.

In cycles I and II, data are analysed as described above in the different studies, focusing on different contexts and levels of the included healthcare organisations, with different professions, interactions, adaptations, influencing factors, and consequences, in each of the specific areas of interest (ie, sustainable working conditions, ethics and patient safety) and phases of the pandemic and aftermath. Findings from cycle I and II (microlevels and mesolevels) are then, in cycle III, complemented with additional data from participants and documents (mesolevels and macrolevels), and analysed cross-sectionally, considering the different system levels and phases of the pandemic, to develop further the understanding of the complexity, including interactions and dynamic adoptions that been made in the included healthcare organisations, influencing factors within the organisations, influencing factors from the macrolevel, and the consequences on working conditions, ethics and patient safety. Finally, in cycle IV, the findings from all the previous cycles are supplemented with knowledge from the current literature to provide an empirical and theoretical foundation for developing a substantive theory of resilient performance during the pandemic rampage. Hence, the cycles of the project are closely integrated and, although not linear, built on each other and enable theory to evolve.

The development of theoretical understanding is a process between the researcher and the empirical data, and the voice of the participants is rendered into the findings. The level of abstraction/theorisation in the studies depends on the content and depth of the data obtained. Hence, the research team is open to settling with a lower level of abstraction, such as purely descriptive results or conceptual modelling in some studies. These results can also be of substantial usefulness. The most important quality criterion in grounded theory is that the findings are truly grounded in the data. The findings should 'fit' the empirical situations in the social context under study, 'work' to explain what was going on in the studied context and have 'relevance' by representing subjects of real concern to the people involved. Finally, the developed theory or conceptual model should have 'modifiability', meaning that as changes take place in reality and studies are performed in other contexts, the theory can evolve.[33 38]

Based on the philosophical orientation of the project, reality is multiple, complex, socially constructed, subjectively perceived, and can be interpreted but not fully known. A further assumption is that knowledge is created within the interaction between the researchers and the participants. Consequently, objectivity is not desirable. However, to ensure that the findings are grounded in the empirical data, the researchers have to be reflective regarding their own preunderstanding.[39] Strategies appropriate for qualitative studies are applied to achieve trustworthiness.[40] The primary analysis in each study will be conducted by two researchers, and reflective discussions will take place with other members of the research team. Clinical experiences within the research team enable theoretical sensibility. Credibility is enhanced through an audit trail, collaboration in the analysis, comprehensive memo writing, and reflective discussions within the research group, which include also individuals without experience of care during the pandemic, an audit trail, peer-reviews and in-depth methodological descriptions. The transferability of the findings will be enhanced by in-depth contextual descriptions.

### Patient and public involvement

There was no patient or public involvement in the design and conduct of this study.

## ETHICS AND DISSEMINATION

The project was approved by the Swedish Ethical Review Authority (Ref. No. 2020-04187). The continuing effect of the pandemic on healthcare means that the researchers have to be responsive to the participants' conditions, both regarding time and emotions. The participants' voluntariness and possibility of withdrawing without consequences are emphasised in the information letter, as well as in the interview situation. Regardless of whether the requested person consented, occupational health services were available for those who experienced emotional discomfort. All data are handled in accordance with the General Data Protection Regulation (EU 2016/679) and stored in a secured place that is available only to the research team; the data will be retained for ten years and then disposed of.

Brief feedback regarding the findings will continuously be provided to the included healthcare organisations to enable pressing improvement work. Academic dissemination of the findings will occur through publication in peer-reviewed journals, following the Consolidated Criteria for Reporting Qualitative Research[41] or equivalent guidelines. The findings will also be presented at academic and practitioner conferences within the area of quality and safety in healthcare.

## DISCUSSION

This multilevel project aims to improve the understanding of how healthcare adapted during the COVID-19 pandemic, the processes involved and the consequences the situations has had on working conditions, ethics and patient safety. A complex adaptive system, such as healthcare, contains many different interacting elements and constantly interacts with the environment.[1 16 17] A complex system's ability to self-organise contributes to its stability and resilience during most disturbances. The pandemic, however, demonstrates how changing dynamics have the potential to challenge and to destabilise the system.[1]

Understanding flexibility and adaptive capacities is a main concern for the field of resilience in healthcare.[19 20] Hence, the ongoing pandemic provides unique possibilities to study system-wide resilient performance in real time.

Resilient healthcare is a growing research area,[31 42 43] and theorisation has enabled, for example, the integrated resilience attribute framework,[16] to be developed. However, there is still a need to understand further the interplay between different system levels.[15 16 20 30] Additionally, knowledge about a crisis with the magnitude of the COVID-19 pandemic is lacking. This situation altogether advocates an exploratory research design with openness for the empirical data and main concern of the participants. The current project is thereby underpinned by the methodological assumptions of grounded theory,[32 33 39] which emphasises theorisation built on empiricism. With this method, a flexible yet systematic approach is applied, with openness for what is discovered. The approach enables ideas and concerns that emerge in the early stage of the research to be handled and to be explored further later in the research trajectory. With emphasis on understanding processes, grounded theory[39] provides tools with which to explore complex and continuously changing social contexts and to develop conceptual models or substantive/mid-range theories that can explain the phenomena under study.

When researching the phenomenon of resilience, some core elements should be addressed, namely, the purpose of resilience (resilience for what?), the triggers that activate eventual latent capabilities (resilience to what?), the resources involved (resilience of what?) and the processes supporting resilience (resilience through what?).[21] These core elements align with probing questions used to facilitate analysing processes in grounded theory, namely, what is going on, when, with whom and how,[33] in the search for conditions, actions/interactions/reactions and consequences. The present project explores resilience for the purpose of providing high-quality care. To determine what triggers and activates resilient performance in healthcare during the pandemic, the conditions over time in different contexts and processes that support resilience, adaptive actions, interactions and reactions will be identified and explored. Involved resources (internal and external) will also be identified by exploring different enrolled actors, elements and affecting factors at different system levels. Finally, as resilience per se is not normative,[20 22] consequences experienced regarding working conditions, ethics and patient safety will be explored as indicators for the ability to provide high-quality care. The described approach may seem sequential; however, the research process is iterative and dynamic.

The project has limitations that should be acknowledged. First, the studies are performed in the healthcare organisations in two Swedish regions only. However, these regions were heavily affected by the pandemic, and transferability of the findings will be enhanced by in-depth contextual descriptions. Second, although complemented with internal documents from the healthcare organisations, and official national and international documents, data are mainly gathered from staff. The first-hand perspective of other stakeholders, such as patients and their families, authorities, financiers and politicians, are not included. Further studies should explore the phenomenon in other settings and also preferably include other stakeholders. Third, the project is inevitably affected by the progress of the COVID-19 pandemic and the staff's ability to participate; therefore, the research trajectory is difficult to predict precisely.

To our knowledge, this is the first study using grounded theory[33] to study working conditions, ethics and patient safety during a healthcare crisis through a lens of resilience. The approach enables the development of theoretical understanding from empirical data. Additionally, when developing theory, the empirical data will be supplemented with knowledge from the current literature to provide both empirical and theoretical foundations for the development of a substantive theory of resilient performance during the pandemic's rampage. The applied design makes the voice of the people involved in healthcare heard and is anticipated to generate in-depth knowledge that can be used to better understand adaptive capacity and the nature and context of adaptations, including the interplay between different system levels, success factors for resilient performance, influencing factors and consequences for sustainable work environment, ethical approach and patient safety. This project can create an important basis for designing interventions with a focus on preparedness to manage current and future challenges in healthcare.

**Author affiliations**
[1] Department of Public Health and Caring Sciences, Uppsala University, Uppsala, Sweden
[2] Centre for Clinical Research, Uppsala University, Region Sörmland, Eskilstuna, Sweden
[3] Department of Anesthesia and Intensive Care Unit, Falun Hospital, Region Dalarna, Falun, Sweden
[4] School of Health and Welfare, Dalarna University, Falun, Sweden
[5] Centre for Clinical Research, Uppsala University, Region Västmanland, Västerås, Sweden
[6] Department of Emergency Medicine, Falun Hospital, Region Dalarna, Falun, Sweden
[7] CLINTEC, Karolinska Institute, Stockholm, Sweden
[8] Department of Health and Caring Sciences, Linneuniversitet, Kalmar/Växjö, Sweden
[9] LIME, Karolinska Institutet, Stockholm, Sweden

**Acknowledgements** The authors would like to acknowledge the healthcare professionals who critically reviewed and gave feedback on the topic guide before the data collection started. We also want to acknowledge all staff and managers in the healthcare organisations that so far contributed to the study by telling their stories and providing the research team with documents to promote our understanding of their situations.

**Contributors** PB-S was responsible for the study design, the protocol draft and the final version. CG contributed to the study design and drafted substantial parts of the manuscript. ML-K, LN and ME provided intellectual input into the study design and contributed to critical review and substantial editing of the draft. A-SK, MC, ECM and MH contributed to critical review and editing of the draft. All authors approved the final version of the manuscript.

**Funding** This work was supported by the Centre of Clinical Research Sörmland, Uppsala University, Sweden (Grant No. DLL-940876), and the Regional Research Council in Mid Sweden (Grant No. RFR-939378). Additionally, Region Västmanland, Region Sörmland, Region Dalarna and Dalarna University, Sweden, support the project with in-kind funding.

**Competing interests** None declared.

**Patient and public involvement** Patients and/or the public were not involved in the design, or conduct, or reporting, or dissemination plans of this research.

**Patient consent for publication** Not applicable.

**Provenance and peer review** Not commissioned; externally peer reviewed.

**ORCID iDs**
Petronella Bjurling-Sjöberg http://orcid.org/0000-0003-0245-3057
Camilla Göras http://orcid.org/0000-0002-0883-4072
Malin Lohela-Karlsson http://orcid.org/0000-0002-9376-846X
Lena Nordgren http://orcid.org/0000-0003-0667-7111
Ann-Sofie Källberg http://orcid.org/0000-0002-0681-9768
Emelie Condén Mellgren http://orcid.org/0000-0003-4525-1623
Mats Holmberg http://orcid.org/0000-0003-1878-0992
Mirjam Ekstedt http://orcid.org/0000-0002-4108-391X

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
