## [Reviewer comments · BMJ Open]

ARTICLE DETAILS

TITLE (PROVISIONAL)	Resilient performance in healthcare during the COVID-19 pandemic (ResCOV): study protocol for a multilevel grounded theory study on adaptations, working conditions, ethics and patient safety.
AUTHORS	Bjurling-Sjöberg, Petronella; Göras, Camilla; Lohela-Karlsson, Malin; Nordgren, Lena; Källberg, Ann-Sofie; Castegren, Markus; Condén Mellgren, Emelie; Holmberg, Mats; Ekstedt, Mirjam

VERSION 1 – REVIEW

REVIEWER	Siri Wiig University of Stavanger, Faculty of Health Sciences
REVIEW RETURNED	03-Jul-2021

GENERAL COMMENTS	Dear authors, thank you for the opportunity to review your protocol. This is a very well written protocol. It is an innovative and important study. We will learn a lot from this work and I look forward to see the results. I only have some minor revisions I would suggest for the authors to improve the protocol for clarity: 1. Describe with a bit more detail how the macro level is included, what organisations are included in your operationalisation of the macro level and how are they recruited.2) include the dates for the study (start and end)3) Explain a bit more details on integration of data from the different levels4) Include info on dissemination in your abstract5) Consider to include research questions that are linked to your aims6) I suggest to include some more limitations to your study. I wish the authors good luck!
---

REVIEWER	Andrew Johnson Townsville Hospital and Health Service
REVIEW RETURNED	30-Sep-2021

GENERAL COMMENTS	This is a really important and timely piece of work, well considered and appropriately referenced. The study reflects a need in the healthcare community to better understand Resilient Health Care in the context of COVID-19 and I congratulate you on a study design that should allow exploration of the dynamic complexity of the evolution of this pandemic and the responses of the humans and the systems exploring the inter-relatedness across system levels. I look forward to the results in due course
--

VERSION 1 – AUTHOR RESPONSE

Reviewer: 1	
Prof. Siri Wiig, University of Stavanger Comments to the Author:	
Dear authors, thank you for the opportunity to review your protocol. This is a very well written protocol. It is an innovative and important study. We will learn a lot from this work and I look forward to see the results. I only have some minor revisions I would suggest for the authors to improve the protocol for clarity:	Thank you for these encouraging words and the perceptive suggestions for improvements. Please see our response point by point:
1) Describe with a bit more detail how the macro level is included, what organisations are included in your operationalisation of the macro level and how are they recruited.	Thank you for illuminating this deficiency. We have in the method section revised the text about sampling and participants (page 17 line 22-30, page 18 line 11-38), and data collection (page 18 line 53-57, page 20 line 3-13) to more clearly describe the levels, and that participants are recruited from the included healthcare organizations and that data from the macro-level consist of applicable official documents.
2) include the dates for the study (start and end)	Thank you for demonstrating that we have been unclear in this matter. The start date of the study was, as described in the manuscript, September 2020. We initially intend to study the complete pandemic but as it is still ongoing, we settle with the first three waves. Hence, the main data gathering will be completed during 2022. However, as additional data gathering might be needed in the theoretical sampling process we are not able to give a definitive date for the end of the study. We have however added some sentences to clarify this in the 'Overall design' section and also added an estimated date for ending the study (page 14, line 54 to page 15, line 3).
3) Explain a bit more details on integration of data from the different levels	We have followed your advice and described this process in more details (page 20, line 30-32 and page 20, line 52 to page 21, line 45).
4) Include info on dissemination in your abstract	We have briefly included this information in the abstract (considering word limitation).
5) Consider to include research questions that are	Thank you for raising this matter. We have

linked to your aims	considered your suggestion and outlined the research questions linked to the objectives (page 13-14).
6) I suggest to include some more limitations to your study.	Thank you for the suggestion. We have added some more limitations in the discussion (page 13 line15- page 14 line 31) and also revised the bullet points under 'Strengths and limitations of this study' (page 8).
Reviewer: 2	
Prof. Andrew Johnson, Townsville Hospital and Health Service Comments to the Author:	
This is a really important and timely piece of work, well considered and appropriately referenced. The study reflects a need in the healthcare community to better understand Resilient Health Care in the context of COVID-19 and I congratulate you on a study design that should allow exploration of the dynamic complexity of the evolution of this pandemic and the responses of the humans and the systems exploring the inter-relatedness across system levels. I look forward to the results in due course	Thank you for these encouraging words